# Estimating Disability-Adjusted Life Years (DALYs) in Community Cases of Norovirus in England

**DOI:** 10.3390/v11020184

**Published:** 2019-02-21

**Authors:** John P. Harris, Miren Iturriza-Gomara, Sarah J. O’Brien

**Affiliations:** 1Institute of Population Health Sciences, University of Liverpool, Liverpool L69 3GL, UK; sjobrien@liverpool.ac.uk; 2National Institute for Health Research (NIHR), Health Protection Research Unit (HPRU) in Gastrointestinal Infections, Liverpool L69 3GL, UK; miren@liverpool.ac.uk; 3Institute of Infection and Global Health, University of Liverpool, Liverpool L69 7BE, UK; 4Modelling, Evidence and Policy Research Group, School of Natural and Environmental Sciences, Newcastle University, Newcastle upon Tyne NE1 7RU, UK

**Keywords:** norovirus, gastrointestinal infections, calicivirus, infectious diseases

## Abstract

Disability adjusted life years (DALYs) have been used since the 1990s. It is a composite measure of years of life lost with years lived with disability. Essentially, one DALY is the equivalent of a year of healthy life lost if a person had not experienced disease. Norovirus is the most common cause of gastrointestinal diseases worldwide. Norovirus activity varies from one season to the next for reasons not fully explained. Infection with norovirus is generally not severe, and is normally characterized as mild and self-limiting with no long-term sequelae. In this study, we model a range of estimates of DALYs for community cases of norovirus in England and Wales. We estimated a range of DALYs for norovirus to account for mixing of the severity of disease and the range of length of illness experienced by infected people. Our estimates were between 1159 and 4283 DALYs per year, or 0.3–1.2 years of healthy life lost per thousand cases of norovirus. These estimates provide evidence that norovirus leads to a considerable level of ill health in England and Wales. This information will be helpful should candidate norovirus vaccines become available in the future.

## 1. Introduction

The concept of the disability-adjusted life year (DALY) has been around since the 1990s. It is a composite measure comprising an estimate of years of life lost with years lived with disability for any disease. The utility of the measure is in providing an estimate of the number of years of healthy life lost because of a disease [1,2], and also allows for comparisons of the burden of illness experienced from different diseases. Essentially, one DALY is the equivalent of a year of healthy life lost if a person had not experienced the disease. Studies have estimated the number of DALYs for diarrheal diseases [3,4,5] and one international study of the burden of disease showed that norovirus led to the highest number of foodborne illnesses [6].

Norovirus is the most common cause of gastrointestinal diseases worldwide. Norovirus activity varies from one season to the next for reasons that are not fully explained. For example, higher activity some years coincided with newly emerging strains [7,8,9], with previous high activity leading to subsequent increases in immunity and associations with climactic conditions, with colder winters possibly increasing transmission [10].

Infection with norovirus is generally not severe. The disease is normally characterized as mild and self-limiting, and having no long-term sequelae. Symptoms of norovirus infections last between one and three days [11,12]. However, recent re-analysis of data from the Second Study of Infectious Intestinal Disease in the Community (IID2 Study) suggests that the illness lasts longer than this, particularly in the very young (those aged <5 years) [13]. Furthermore, for some, e.g., those who are hospitalized, a longer periods of symptoms are experienced [14] and, although not necessarily directly causal, there is a measurable level of mortality, particularly in the elderly (those aged >65 years) [15].

Understanding the burden of disease is an important step in developing policies aimed at mitigating or preventing illness. Currently, there are candidate vaccines that show promising results in protection against norovirus disease [16,17,18]. Before deciding to introduce vaccines for disease prevention strategies, a useful estimate of the likely amount of disease it would prevent is an essential measure. Given the variation in norovirus activity from one year to the next, and the variation in the severity of illness experienced by those who are ill with norovirus, a single DALY calculation for norovirus is unlikely to be appropriate or a realistic measure of the burden of the disease.

In this study, we model a range of estimates of DALYs for community cases of norovirus in England and Wales. These ranges of DALYs are achieved by varying the proportions of those experiencing mild and moderate disease, and through varying the range of lengths of illness experienced in mild and moderate cases.

## 2. Methods

### 2.1. Model

The calculation of DALYs is the addition of years of life lost (YLL) + years lost due to disability (YLD). YLL = N × L, where N = number of deaths, L = standard life expectancy at age of death in years; and YLD = I × DW × L, where I = number of incident cases, DW = disability weight, and L = average duration of the case until remission or death.

### 2.2. Disease Weightings

As there are no disease weightings specific to norovirus, these were taken from the Global Burden of Diseases measures for mild and moderate diarrhea, which is 0.061 for mild and 0.202 for moderate diarrhea, respectively [19]. The proportions of mild and moderate cases varied from 50% to 95% with the length of illness, which was between 1 to 1.5 days for mild cases, and 2 to 3 days for moderate cases. The incidence of norovirus was taken from the updated measures previously applied to the infectious intestinal disease (IID2) study of 59/1000 population [20].

The estimates were derived as follows: Mild disease I × (proportion mild) × DW × L, where mild proportion varied between 0.5 to 0.95, and L varied from 1 to 2 days; and Moderate disease I × (proportion moderate) × DW × L, where proportion moderate varied between 0.5 to 0.95 and L varied from 2.5 to 3.5 days. These provided matrices of results for mild and moderate that were combined to provide a full matrix of estimated DALYs.

The upper and lower estimates were calculated using the lower and upper estimates of the disease weightings from the Global Burden of Diseases (GBD) study (mild diarrhea 0.036–0.093, and moderate diarrhea 0.133–299).

### 2.3. Data Sources

#### Deaths

We used the Office for National Statistics (ONS) data on the principle cause of deaths and the years of life lost to estimate the levels of mortality and attribute years of life lost to the number of deaths at each age. Deaths attributed to norovirus were taken from ONS data from 2013 to 2016 as these were the years where the data were available for electronic download (https://www.nomisweb.co.uk/home). The ICD10 code A08.1 Acute gastroenteropathy due to Norwalk agent was used to obtain the number of deaths in each age group.

The number of deaths were summed in each year for each age group (in five-year bands) and were averaged from these years. Years of life lost were calculated from life table cohorts from the same years as the ONS data. As the ages for deaths were grouped in five-year age bands, years of life lost were applied at the lowest age within the age group. For example, for a death occurring at age 54, the years of life lost were applied to a fifty-year-old. The average number of years of life lost were multiplied by the average number of deaths for each age group and summed over the year to obtain a total estimate of the years of life lost. The population estimate for England and Wales was taken from the ONS mid-year population estimate.

The upper and lower estimates were derived by calculating the lower and upper quartile of years of life lost and were applied to the lower and upper estimate of the incidence. A further analysis was also conducted on the original estimated incidence of norovirus from the IID2 study [21] to give a full range of possible DALYs.

For comparison of DALYs estimated for norovirus, we used the same approach to calculate DALYs for rotavirus prior to the introduction of the rotavirus vaccine. We have not shown the calculations in this paper; the comparison was made in order to give context to the importance of norovirus as an infection with a comparable disease, which has now become a vaccine-preventable disease.

## 3. Results

### 3.1. Deaths Attributed to Norovirus

In four years, 2013–2016, there was a total of 123 deaths attributed to norovirus according to ONS mortality statistics. The highest number of recorded deaths occurred in older ages, 89% of which were in those aged 70 years and over. Annually, the totals were between 23 and 37 deaths. The total number of years of life lost due to norovirus infections was estimated as 374.75 years (106.37–435.74).

### 3.2. Incident Cases

Assuming an incidence of 59/1000 population, this equated to 3,444,492 incident cases of norovirus a year based on the mid-year population estimate from ONS of 58,381,217. The number of DALYs estimated varied depending on both the proportion of cases experiencing mild illness and the number of days that illness was experienced. Table 1 shows the range of estimates for the DALYs calculated. The estimates ranged between 1159 (801–1298) DALYs, where 95% of cases experienced mild symptoms lasting one day and 5% experienced moderate symptoms lasting 2.5 days, to 4283 (3571–4735) DALYs, where 50% experienced mild symptoms lasting 2 days and 50% experienced moderate symptoms for 3.5 days.

Table 2 shows the number of DALYs per one thousand cases of norovirus in each year. The estimates ranged between 0.34 (0.23–0.38) to 1.24 (1.00–1.33), depending on the proportion of cases experiencing mild and moderate symptoms.

## 4. Discussion

Assessing the incidence of norovirus is challenging, particularly in community cases, because the vast majority of those infected have no contact with medical services and will not have a microbiologically confirmed diagnosis. We have used the re-evaluated incidence estimate previously published from a large prospective study of infectious intestinal disease (IID2) to estimate the number of incident cases. This measure was used because it is felt to be a better reflection of the incidence as the original study used a more rigid criterion for determining norovirus infection than is routinely used in laboratories [20].

In this study, we have estimated a range of disability-adjusted life years for norovirus to account for mixing of the severity of disease, and to account for the range of the lengths of illness suffered by those who are infected. The range of DALYs estimated was between 1159 and 4283 per year, and between 0.3 and 1.2 years of healthy life are lost per thousand cases of norovirus. These estimates provide evidence that, despite infection with norovirus being considered as a generally mild, self-limiting illness with no long-term sequelae, the prodigious number of cases experienced each year, almost four million cases, leads to a considerable level of ill health in England and Wales.

The contribution of mortality to DALYs for norovirus infection is low. This is not unexpected given that mortality is largely associated with illness in elderly patients [15], particularly in developed countries and, therefore, contribute only a small amount to DALYs as a whole. This situation is likely to be different in less developed countries where mortality through diarrheal disease is greater, and particularly in young children, where their mortality will contribute more to DALYs. This is an important point in light of the introduction of a vaccine for rotavirus. Norovirus now plays a greater role in contributing to gastrointestinal disease in developing countries. Norovirus affects all ages but has been shown, even in developed countries, to have a higher rate in children than other age groups [20,22].

The estimates for DALYs in this study are in line with a study published in Australia that estimated 0.5 DALYs per thousand cases for norovirus infections [23], and a large study in the United States that estimated 9900 DALYs for norovirus [5] with 5.46 million incident cases equating to 1.8 DALYs per 1000 cases. Our estimate was also similar to a recent study in the Netherlands that estimated 1647 DALYs per year at 0.3 per 100 infections [24], and 1754 DALYs at 0.2 per 100 cases [24].

In recent times, a vaccine has been introduced for rotavirus infection in many countries. The DALY estimates for rotavirus gastrointestinal disease have been estimated as similar to those published here for norovirus at 1820 DALYs per year [4]. Using the same approach we used for calculating DALYs for norovirus, we calculated DALYs for rotavirus. We estimated that, for rotavirus, the DALYs were between 670 and 1216 DALYs per year. For this calculation, we used the incidence of rotavirus infections estimated from the IID2 study. As there was only one death attributed to rotavirus for the period of the study according to ONS, we used mortality associated with rotavirus estimated from a modeling study [25]. The comparison was useful in that the estimates are for DALYs in years, prior to the introduction of the vaccine, and provide a useful comparison from this perspective. The introduction of the rotavirus vaccine has seen a considerable reduction in illness and £12.5 million reduction in healthcare costs [26].

There are few studies estimating the impact of norovirus and providing an estimate of the DALYs and, to our knowledge, this is the only one that provides this estimate from the United Kingdom. This information, and the comparison with DALYs attributed to a viral gastrointestinal pathogen where vaccines have been introduced, will be helpful for guiding policy makers, should the candidate norovirus vaccines be as effective and become available in the near future.

## Figures and Tables

**Table 1 viruses-11-00184-t001:** Estimates of disability-adjusted life years (DALYs) for norovirus infections with an annual incidence of 59/1000 population.

	Percent with Mild Symptoms			
**Length of Illness (Mild/Moderate)**	95	80	65	50
1/2.5 days	1159.36 (801.89–1298.81)	1787.43 (1358.63–1989.69)	2415.51 (1915.38–2680.57)	3043.58 (2472.13–3371.45)
1.5/2.5 days	1432.61 (1044.10–1599.39)	2017.54 (1562.60–2242.81)	2602.47 (2081.11–2886.23)	3187.39 (2599.61–3529.65)
2/2.5 days	1705.86 (1286.32–1899.96)	2247.64 (1766.58–2495.92)	2789.43 (2246.84–3091.88)	3331.21 (2727.09–3687.84)
1/3 days	1206.99 (844.10–1351.20)	1977.93 (1527.49–2199.24)	2748.87 (2210.89–3047.28)	3519.82 (2894.29–3895.32)
1.5/3 days	1480.24 (1086.32–1651.77)	2208.04 (1731.47–2452.35)	2935.83 (2376.62–3252.93)	3663.63 (3021.77–4053.51)
2/3 days	1753.48 (1328.54–1952.35)	2438.14 (1935.44–2705.47)	3122.79 (2542.35–3458.59)	3807.45 (3149.25–4211.71)
1/3.5 days	1254.61 (886.32–1403.59)	2168.43 (1696.36–2408.79)	3082.24 (2506.40–3413.98)	3996.06 (3316.44–4419.18)
1.5/3.5 days	1527.86 (1128.53–1704.16)	2398.53 (1900.33–2661.90)	3269.20 (2672.13–3619.64)	4139.87 (3443.93–4577.38)
2/3.5 days	1801.11 (1370.75–2004.74)	2628.64 (2104.31–2915.01)	3456.16 (2837.86–3825.29)	4283.69 (3571.41–4735.57)

Note: Figures in brackets are upper and lower estimates.

**Table 2 viruses-11-00184-t002:** Estimated DALYs per 1000 cases of norovirus.

	Percent with Mild Symptoms			
Length of Illness (Mild/Moderate)	95	80	65	50
1/2.5 days	0.34 (0.23–0.38)	0.52 (0.39–0.58)	0.70 (0.56–0.78)	0.88 (0.72–0.98)
1.5/2.5 days	0.42 (0.30–0.46)	0.59 (0.45–0.65)	0.76 (0.60–0.84)	0.93 (0.75–1.02)
2/2.5 days	0.50 (0.37–0.55)	0.65 (0.51–0.72)	0.81 (0.65–0.90)	0.97 (0.79–1.07)
1/3 days	0.35 (0.25–0.39)	0.57 (0.44–0.64)	0.80 (0.64–0.88)	1.02 (0.84–1.13)
1.5/3 days	0.43 (0.32–0.48)	0.64 (0.50–0.71)	0.85 (0.69–0.94)	1.06 (0.88–1.18)
2/3 days	0.51 (0.39–0.57)	0.71 (0.56–0.79)	0.91 (0.74–1.00)	1.11 (0.91–1.22)
1/3.5 days	0.36 (0.26–0.41)	0.63 (0.49–0.70)	0.89 (0.73–0.99)	1.16 (0.96–1.28)
1.5/3.5 days	0.44 (0.33–0.49)	0.70 (0.55–0.77)	0.95 (0.78–1.05)	1.20 (1.00–1.33)
2/3.5 days	0.52 (0.4–0.58)	0.76 (0.61–0.85)	1 (0.82–1.11)	1.24 (1.04–1.37)

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
