# Peer review of "Estimating Disability-Adjusted Life Years (DALYs) in Community Cases of Norovirus in England"

_viruses, 2019, doi:10.3390/v11020184_

Reviewer 1 Report

The manuscript by Harris et al applies the concept of Disability Adjusted Life Years to community cases of norovirus in England. Introduction and Materials are well described. Conclusions are supported by the results. Overall, the manuscript is well organized and easy to read. Minor comments are described below.

The author’s line should be fixed.

Introduction: Line 41-42: “…..between one and three days.” Needs a reference

Discussion: Line 155-159: The 3 studies mentioned here need references.

Line 168-169: “The comparison was…”. This sentence could be deleted. Instead, this paragraph could mention the change in DALYs for rotavirus after vaccine introduction, and what it could be hope for norovirus.

Author Response

We thank the reviewer for their helpful comments.

The author's line has now had the erroneous 'and' removed.

Introduction: Line 41-42: “…..between one and three days.” Needs a reference

We have now referenced this. We have also added a reference for the statement on lines 43-44 which was originally referenced as unpublished data

Discussion: Line 155-159: The 3 studies mentioned here need references

We have now added references to those studies.

Line 168-169: “The comparison was…”. This sentence could be deleted. Instead, this paragraph could mention the change in DALYs for rotavirus after vaccine introduction, and what it could be hope for norovirus.

We have used the comparison of rotavirus for the period prior to the introduction of the vaccine for a comparison of DALYs for a similar type of illness.  We give this comparison as it gives a useful baseline for policymakers should an effective vaccine for norovirus become available.  At the moment there is no data the authors are aware of that estimates DALYs for rotavirus in the UK post introduction of the vaccine.  We have however, based on this comment added the following sentence: "The introduction of the rotavirus vaccine has seen a considerable reduction in illness and £12.5 million reduction in healthcare costs" and referenced accordingly.

Reviewer 2 Report

Discussion points:

There are other, more recent data sets or estimates available for other countries (particularly SurvSat data from Germany), it would be of interest to compare to these.

Could the rotavirus calculation be included in some other form, for example supplementary information. To allude to it without including the details is a shame.

Asymptomatic infection is not mentioned. This is a large loss of information as it is estimated to be as high as 30%. This may affect the proportions of mild or moderate symptoms and could be mentioned in the discussion.

Similarly, how does this study account for misdiagnosis.

There has been shown to be a disparity in reporting between age groups ie. reporting is much better for the elderly and young children. How may this affect the estimates of DALYs.

Deaths are assessed from 2013 to 2016- is this time span sufficient as it includes a pandemic year?

The duration of infection can be much higher for immuno-compromised individuals, how do the authors address this in these results?

Author Response

There are other, more recent data sets or estimates available for other countries (particularly SurvSat data from Germany), it would be of interest to compare to these.

We have not found the SurvSat data and have identified one reference which is slightly newer but similar to the already referenced article from the Netherlands. We have now added a reference to this study and added it in the discussion.

Could the rotavirus calculation be included in some other form, for example supplementary information. To allude to it without including the details is a shame.

We feel that the rotavirus data are a useful comparison but not the main focus of the paper.  We calculated DALYs estimates in exactly the same way, whilst acknowledging we could not use mortality data from ONS for rotavirus owing to the lack of it, but the utility of the rotavirus estimate was as a comparison of a diarrhoeal disease similar to norovirus and having similar DALYs and for which a vaccine has been introduced.  

Asymptomatic infection is not mentioned. This is a large loss of information as it is estimated to be as high as 30%. This may affect the proportions of mild or moderate symptoms and could be mentioned in the discussion.

We agree that the question around asymptomatic infection is one of importance, particularly for transmission.  However, for this study DALYs are concerned with the effect of symptomatic illness.  In this instance infection without symptoms would not lead to a measurable level of disease and one that would lead to time off sick.  Having said that we have estimated a range of plausible estimates based on differing proportions of mild illness experienced so we have, we feel dealt with the issue of high levels of mild diseases in the models we present.

Similarly, how does this study account for misdiagnosis.

The DALYs estimates are based on incidence of norovirus disease in the community which we have gathered from estimates taken from a large cohort study of infectious intestinal disease in the UK.  This study estimated the incidence of norovirus illness in the community.  The study estimates are based on diagnosis of norovirus using PCR for cases presenting to their GP or volunteers who report illness providing stool specimens.  The study is referenced in the paper  

There has been shown to be a disparity in reporting between age groups ie. reporting is much better for the elderly and young children. How may this affect the estimates of DALYs.

We agree that there are differences in the incidence in age groups.  However, in this study we have used the incidence estimated for all ages from a large cohort study in the UK which accounted for the differences in rates of disease by age.   

Deaths are assessed from 2013 to 2016- is this time span sufficient as it includes a pandemic year?

We used the deaths estimates for these years as they are those available where norovirus is specifically mentioned in the cause of death.  Whilst some parts of the world experienced high levels of norovirus infections during this time, the UK has not seen any abnormal activity.  In fact data from Public Health England suggests for this period norovirus activity has been largely very calm and at levels lower than the previous five years.   

The duration of infection can be much higher for immuno-compromised individuals, how do the authors address this in these results?

We agree that norovirus infection in immuno-compromised individuals is a problem particularly as it leads to long term shedding.  However, this study is based on the incidence in the community and given that most cases of immuno-compromised individuals are likely to be hospital based and suffering from serious illness it would be difficult to attribute DALYs to norovirus as the underlying disease causing the immuno-compromised state would be the main reason for the incapacity.  Norovirus in this instance is incident to rather than the cause of the DALYs.